# Mechanism of polyadenylation-independent RNA polymerase II termination

Srinivasan Rengachari ✉, Thomas Hainthaler, Christiane Oberthuer,
Michael Lidschreiber & Patrick Cramer ✉

The mechanisms underlying the initiation and elongation of RNA polymerase II (Pol II) transcription are well-studied, whereas termination remains poorly understood. Here we analyze the mechanism of polyadenylation-independent Pol II termination mediated by the yeast Sen1 helicase. Cryo-electron microscopy structures of two pretermination intermediates show that Sen1 binds to Pol II and uses its adenosine triphosphatase activity to pull on exiting RNA in the 5′ direction. This is predicted to push Pol II forward, induce an unstable hypertranslocated state and destabilize the transcription bubble, thereby facilitating termination. This mechanism of transcription termination may be widely used because it is conceptually conserved in the bacterial transcription system.

The transcription cycle consists of three phases—initiation, elongation and termination of the pre-mRNA chain synthesis[1]. Based on a large number of published studies, we have a detailed understanding of the molecular mechanisms underlying RNA polymerase II (Pol II) initiation and elongation[1–3]. In contrast, the mechanisms of Pol II termination remain poorly understood at a structural level. Termination determines intergenic boundaries and prevents pervasive transcription[4,5]. When termination is compromised, it leads to the synthesis of cryptic RNA transcripts that can be cytotoxic[6]. Defective termination also potentiates unscheduled Pol II stalling, which causes conflicts with other RNA and DNA polymerases[4,7]. Dysregulation of transcription termination has further been related to viral infection and cancer[8]. Despite its importance, factor-dependent eukaryotic transcription termination has not been studied structurally.

Pol II termination can occur in a polyadenylation site (PAS)-dependent or PAS-independent way. PAS-dependent termination occurs at the downstream end of protein-coding genes[9], is mediated by the 5′->3′ exonuclease Rat1 in yeast (XRN2 in human)[10,11] and uses a so-called torpedo mechanism[12]. In contrast, PAS-independent termination is achieved by the NNS complex in yeast that comprises the subunits Nrd1, Nab3 and Sen1 (refs. 13,14). The NNS complex is used predominantly to terminate Pol II transcription of noncoding RNAs such as small nuclear and nucleolar RNA[15]. It is also used to terminate pre-mRNA transcription in a 'fail-safe' mechanism[16]. Genome-wide perturbation studies showed that the NNS complex globally restricts pervasive noncoding RNA transcription in yeast[17].

The NNS subunits Nrd1 and Nab3 possess RNA-binding activity[18,19], whereas Sen1 contains an adenosine triphosphate (ATP)-dependent 5′->3′ helicase activity[20]. Sen1 belongs to the ancient SF1b superfamily of helicases[21]. It can terminate Pol II in the absence of Nrd1 and Nab3 (ref. 14) to remove stalled Pol II from the genome in response to R-loop formation[22,23]. The function of Sen1 in cotranscriptional R-loop resolution is conserved in its mammalian ortholog SETX (ref. 24). Mutations in *SETX* have been implicated in neurodegenerative diseases[25]. Structural studies of Sen1 are limited to its C-terminal helicase domain, which is sufficient for Pol II termination[26]. The Sen1 structure features an ATPase domain with two 'RecA lobes' (lobes 1 and 2) and additional brace, stalk, β-barrel and prong modules[26]. Despite these studies, it is unknown how Sen1 interacts with the transcription machinery and how it engages with RNA, which are prerequisites for termination[14]. In this work, we report two cryo-electron microscopy (cryo-EM) structures of Sen1-containing Pol II complexes and propose the mechanism of PAS-independent Pol II termination.

## Results

### Preparation of functional Pol II pretermination complex

To elucidate the molecular mechanism of Sen1-dependent Pol II termination, we prepared a functional yeast Pol II pretermination complex

Department of Molecular Biology, Max Planck Institute for Multidisciplinary Sciences, Göttingen, Germany.
✉e-mail: srinivasan.rengachari@mpinat.mpg.de; patrick.cramer@mpinat.mpg.de

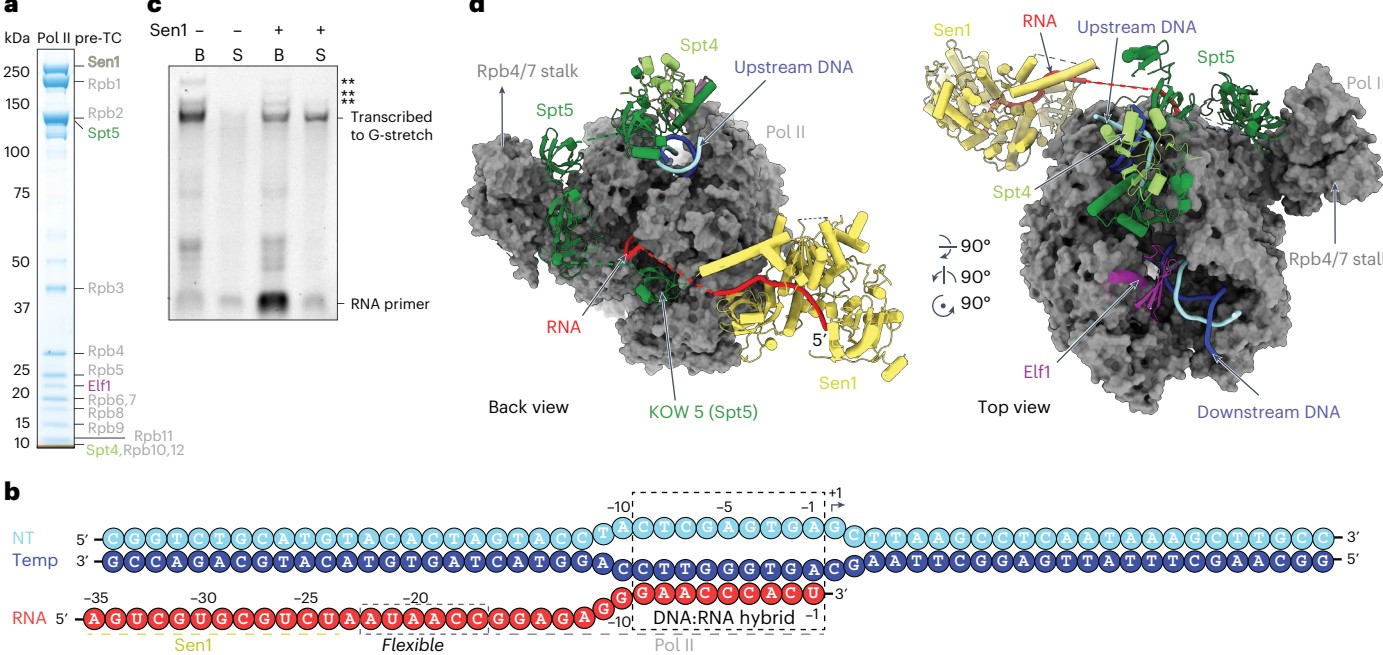

**Fig. 1 | Structure of yeast Pol II pre-TC. a**, SDS–PAGE analysis (replicated a minimum of three times) showing a peak fraction of the pre-TC containing all its components in stoichiometric amounts. The color codes of the subunits are used throughout the paper, unless stated otherwise. **b**, Schematic view of the DNA–RNA hybrid template used for the reconstitution of the pre-TC. The positions of nucleic acids in the pre-TC structure are numbered. Functional parts of the pre-TC that bind to the template are specified with color-coded labels. **c**, Denaturing gel showing termination activity of the pre-TC depicted in **a**. The

final products corresponding to terminated transcripts are marked and labeled. B, bead fraction; S, supernatant fraction. Asterisks indicate the longer transcripts resulting from misincorporation. The experiment was repeated a minimum of three times. **d**, Representative views of the structure of Sen1 bound to Pol II. The relative orientations of the views are indicated by arrows and the individual components of the pre-TC are labeled with color codes. Pol II is depicted in surface representation and the factors Spt4, Spt5, Elf1 and Sen1, along with the nucleic acids, are shown in cartoon representation.

(pre-TC). Since the elongation factors Spt5, Spt6 and Elf1 co-occupy genomic regions with Nrd1 (ref. 27), we reconstituted a minimal pre-TC from cyclin-dependent kinase 7 (CDK7)-phosphorylated Pol II, Spt4, Spt5, Elf1 and Sen1 (Fig. 1a). The pre-TC was assembled on a DNA–RNA template with a 35-nt-long RNA (Fig. 1b and Methods). The pre-TC was resolved using sucrose-gradient ultracentrifugation (Fig. 1a) and was functional in Pol II termination in vitro (Fig. 1c and Supplementary Fig. 1).

For structural studies, we stabilized the pre-TC using GraFix[28] and collected single-particle cryo-EM data (Methods). We obtained a three-dimensional (3D) reconstruction of the pre-TC at a nominal resolution of 2.8 Å from 95,644 particles (Extended Data Figs. 1 and 3 and Supplementary Fig. 2). Densities were observed for all of the pre-TC components (Extended Data Fig. 1). The Sen1-containing region of the map was improved using focused 3D classification and masked refinement, resulting in a local map at 3.3-Å resolution. The atomic model of the pre-TC was built using published structures of Pol II, the DNA–RNA hybrid and Sen1 in the cryo-EM map. For Spt4, Spt5 and Elf1, we combined models from published structures and the Alpha-Fold database (Methods and Table 2)[29]. After manual adjustments and real-space refinement, the final model of the pre-TC showed a good fit to the cryo-EM maps with good stereochemistry (Extended Data Figs. 3 and 4 and Table 1).

## Structure of the Pol II pre-TC

The overall structure of the pre-TC shows a canonical elongation complex (EC) with a 9-bp DNA–RNA hybrid and the Pol II active site in the post-translocated state (Fig. 1b,d and Supplementary Fig. 3). The structure also shows that Sen1 binds to the pre-TC but does not interact with Spt4, Spt5 and Elf1 (Fig. 1d). Instead, Sen1 directly binds the Pol II subunit Rpb3 that is located near the RNA exit site (Fig. 1d). In particular, Sen1 uses its β-barrel module to contact Domain2 of

Rpb3 (Fig. 2a,b). This interaction involves hydrogen bonds between the main-chain atoms of the β6-strand of Rpb3 and the β1-strand of the Sen1 β-barrel module, in addition to hydrophobic, polar and ionic interactions (Fig. 2a). Superposition of our yeast Pol II pre-TC with its elongation counterpart Pol II EC*[30] shows that Sen1 clashes with the Rtf1 subunit of the Paf1 complex (Extended Data Fig. 5). In particular, the brace and prong modules of Sen1 clash with the pincer helices and plus 3 regions of Rtf1, respectively.

The structure of the pre-TC also shows that Sen1 binds the RNA transcript, burying up to 13 nt (register −35 to −23 relative to the Pol II active site) (Figs. 1b and 2b) within its substrate-binding channel. Of the remaining 22 nt, 16 are buried within Pol II (register −16 to −1), whereas the linking six residues (register −22 to −17) were not resolved because of flexibility (Fig. 1b). The structure is consistent with prior work showing that a 30-nt nascent RNA is minimally required for Sen1-dependent termination[14]. Sen1 contacts RNA not only with its ATPase lobes 1 and 2 but also with its β-barrel, stalk and prong modules (Fig. 2b,c).

## Sen1 dynamics in the Pol II pre-TC

To uncover the mechanism of Sen1-mediated Pol II termination, we determined the cryo-EM structure of the pre-TC in the presence of adenosine diphosphate (ADP)·BeF₃, which mimics the transition state during ATP hydrolysis. The final map from 9,095 particles was refined to a nominal resolution of 4.3 Å, with the local map of Sen1 extending to 4.4 Å (Extended Data Figs. 2 and 3 and Supplementary Fig. 2). The slightly lower resolution limited building of the RNA chain within Sen1 to 9 nt (register −34 to −26; Table 2). Compared to the first structure in the apo state, Pol II and elongation factors were unaltered (Extended Data Fig. 6).

Structural superposition of our two pre-TC structures shows that ADP·BeF₃ binding induces a rearrangement of lobe 2 of the ATPase domain of RNA-bound Sen1 (Fig. 3a). Specifically, lobe 2 rotates by

**Table 1 | Cryo-EM data collection and processing information**

| | Pol II pre-TC, overall map | Pol II pre-TC, Sen1–RNA apo local map | Pol II pre-TC with ADP·BeF$_3$, overall map | Pol II pre-TC with ADP·BeF$_3$, Sen1–RNA ADP·BeF$_3$ local map |
|---|---|---|---|---|
| | EMD-19019, PDB8RAM | EMD-19020, PDB8RAN | EMD-19022, PDB8RAP | EMD-19021, PDB8RAO |
| **Data collection and processing** | | | | |
| Magnification | ×81,000 | | ×81,000 | |
| Voltage (kV) | 300 | | 300 | |
| Electron exposure (e⁻ per Å$^2$) | 40.02 | | 39.76 | |
| Defocus range (μm) | −0.5 to −3.0 | | −0.5 to −2.5 | |
| Pixel size (Å) | 1.05 | | 1.05 | |
| Micrographs collected | 48,735 | | 29,640 | |
| Initial particle images (no.) | 9,710,589 | | 4,333,083 | |
| Final particle images (no.) | 95,644 | | 9,095 | 15,633 |
| Map resolution (Å) | 2.8 | 3.25 | 4.3 | 4.4 |
| FSC threshold | 0.143 | | 0.143 | |
| Map resolution range (Å) | 2.5–6.5 | 2.9–6.9 | 3.4–9.2 | 4.0–7.6 |
| Model-to-map fit resolution (Å, FSC > 0.5) | 2.8 | 3.25 | 4.3 | 4.4 |
| **Refinement** | | | | |
| Initial model used (PDB code) | 7NKX, 6I59 | 6I59, 2XZO | 7NKX, 6I59 | 6I59, 2XZO |
| Map sharpening $B$ factor (Å$^2$) | 0 | 0 | 0 | 0 |
| Model composition | | | | |
| Nucleic acids | 94 | 13 | 101 | 9 |
| Protein residues | 5,164 | 690 | 4,714 | 694 |
| Ligands | 10 | 0 | 13 | 3 |
| $B$ factors (Å$^2$) | | | | |
| Nucleic acids | 141.35 | 163.96 | 196.04 | 264.66 |
| Protein residues | 110.75 | 144.83 | 196.32 | 258.13 |
| Ligands | 117.83 | -n.a- | 216.16 | 258.97 |
| Root-mean-square deviations | | | | |
| Bond lengths (Å) | 0.002 | 0.001 | 0.002 | 0.001 |
| Bond angles (°) | 0.382 | 0.341 | 0.379 | 0.393 |
| **Validation** | | | | |
| MolProbity score | 1.28 | 1.13 | 1.31 | 1.16 |
| Clashscore | 5.09 | 3.39 | 5.06 | 3.66 |
| Poor rotamers (%) | 0.04 | 0.00 | 0.00 | 0.00 |
| CaBLAM outliers | 1.57 | 0.88 | 1.52 | 0.88 |
| Cβ outliers | 0.00 | 0.00 | 0.00 | 0.00 |
| Ramachandran plot | | | | |
| Favored (%) | 97.96 | 98.83 | 97.82 | 97.97 |
| Allowed (%) | 2.04 | 1.17 | 2.18 | 2.03 |
| Disallowed (%) | 0.00 | 0.00 | 0.00 | 0.00 |

~20° toward lobe 1 through a hinge movement[31], leading to a closure of the cleft between the two lobes (Fig. 3a,b). As a consequence, lobe 2 residues that contact the RNA backbone move by a single nucleotide in the 3′ direction. For example, the main-chain amide of residue R1753 in helix α28 of lobe 2, which hydrogen bonds to the phosphate between residues G −34 and U −33 in the apo state, moves by one register to contact the phosphate between U −33 and C −32 in the ADP·BeF$_3$-bound state (Figs. 2b and 3b).

Comparison of the two pre-TC states with structural studies of other SF1b helicases[32] shows that Sen1 ATPase activity pulls the RNA chain in the 5′ direction. In particular, upon hydrolysis of ATP and ADP release, lobe 2 of Sen1 would swing back to its position observed in the apo state and pull the RNA strand with it (Fig. 3a,b). This causes translocation of the RNA chain by 1 nt in the 3′-to-5′ direction. Notably, rotation of lobe 2 does not interfere with the Sen1–Pol II interaction, showing that Sen1 activity does not alter Pol II binding. In conclusion, these results indicate that Sen1 ATPase action pulls RNA away from the Pol II active site.

**Mechanistic model of PAS-independent termination**

Based on these results and prior work, a model for the molecular mechanism for Sen1-dependent Pol II termination emerges. First,

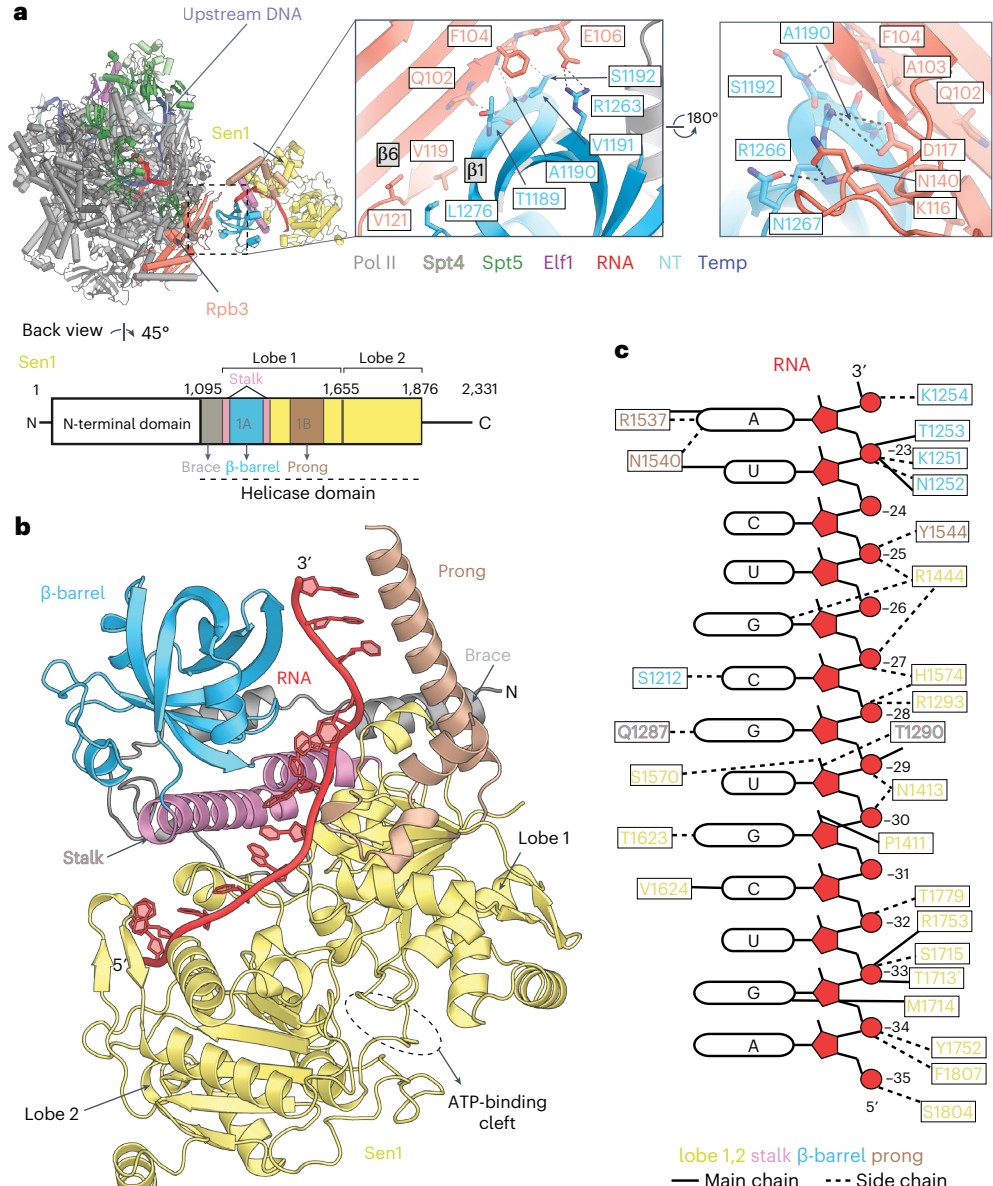

**Fig. 2 | Molecular details of Sen1–Pol II interaction. a**, Transparent cartoon view presenting the details of interaction between Sen1 and Rpb3. The inset of the interaction region is presented as two views. The main-chain and side-chain residues involved in the interaction are indicated with labels and color codes. **b**, Structure of Sen1 bound to the Pol II transcript represented as a cartoon. The color code for the different modules of Sen1 are used as displayed in the domain scheme in **a**. The ATPase lobes and functional modules of Sen1 are labeled with color codes. **c**, Schematic cartoon of the protein–RNA interactions between Sen1 and RNA within the pre-TC. The main-chain and side-chain Sen1 residues within hydrogen-bonding distance to the RNA are indicated as solid and dashed lines, respectively.

Nrd1 and Nab3 bind to nascent RNA and recruit Sen1 (ref. 33). Then, Sen1 translocates along the RNA in the 3′ direction to catch up with elongating Pol II (ref. 34) and dock to Rpb3 (Figs. 2a and 3a). Persistent ATPase activity of Sen1 then pulls the RNA (Fig. 3b) and causes a hypertranslocated state of Pol II that cannot extend the RNA, thereby stalling transcription. The continued pulling on exiting RNA by Sen1 would then shorten the DNA–RNA hybrid and destabilize the transcription bubble, favoring DNA rewinding and duplex reformation. These events lead to displacement of the RNA chain from the Pol II active site followed by the release of nucleic acids from Pol II, thereby causing termination.

This mechanism is consistent with a biochemical study showing that Sen1 can mechanically hypertranslocate stalled Pol II and that intervening with upstream DNA rewinding reduces termination[35].

Another single-molecule study revealed a termination intermediate with partially rewound upstream DNA[36]. An alternative mechanism for helicase-dependent transcription termination[37], the 'allosteric model'[38], is not supported by our data because we do not observe conformational rearrangements in Pol II or its elongation factors upon Sen1 binding (Extended Data Fig. 6). However, we cannot rule out that allosteric changes occur in later steps toward termination. A third mechanism, called 'hybrid shearing', seems to require a strong helicase[39] and is less likely to occur because Sen1 only has weak RNA translocase activity[35].

### Sen1 specificity for RNA polymerases
Apart from Pol II, Sen1 can terminate the yeast Pol I and Pol III (refs. 3,40,41). Analysis of the electrostatic surface potential of the

**Table 2 | Input structural models and model confidence**

| Complex/domain | Chain identifier | Input model (PDB code) | Level of confidence |
|---|---|---|---|
| **Pol II pre-TC** | | | |
| Pol II | A–C, E, F, H–L | 7NKX | Atomic |
| DNA | N, T | 7NKX | Atomic |
| RNA (res: −35 to −26, −9 to −1) | P | 7NKY, 2XZO | Atomic |
| RNA (res:−25 to −23, −16 to −10) | P | 7NKX | Rigid-body fitted |
| Pol II | D, G | 7NKX | Rigid-body fitted |
| Elf1 | M | AlphaFold model | Rigid-body fitted |
| Sen1 | O | 6I59 | Atomic |
| Spt4 | Y | 7NKX | Rigid-body fitted |
| Spt5 (NGN, KOW 1L, 2, 3 and x4) | Z | AlphaFold model | Rigid-body fitted |
| Spt5 KOW 5 | Z | 7NKX | Atomic |
| **Pol II pre-TC ADP·BeF$_3$** | | | |
| Pol II | A–C, E, F, H–L | 7NKX | Atomic |
| DNA | N, T | 7NKX | Atomic |
| RNA (res: −9 to −1) | P | 7NKX, 2XZO | Atomic |
| RNA (res: −34 to −26, −16 to −10) | P | 7NKX | Rigid-body fitted |
| Pol II | D, G | 7NKX | Rigid-body fitted |
| Elf1 | M | AlphaFold model | Rigid-body fitted |
| Sen1 | O | 6I59 | Rigid-body fitted |
| Spt4 | Y | AlphaFold model | Rigid-body fitted |
| Spt5 (NGN, KOW 1L, 2, 3 and x4) | Z | AlphaFold model | Rigid-body fitted |
| Spt5 KOW 5 | Z | 7NKX | Atomic |

pre-TC shows that, in addition to the main-chain and side-chain contacts, the Sen1 β-barrel module and domain 2 of Rpb3 show complementary negatively and positively charged surfaces, respectively (Fig. 3c and Extended Data Fig. 7a). Similarly, Domain 2 of subunit AC40, the Rpb3 counterpart in Pol I and Pol III, also contains a negatively charged surface (Fig. 3c and Extended Data Fig. 7b). Superposition of our pre-TC structures onto Pol I and Pol III suggests that Sen1 can bind AC40 Domain 2 (Extended Data Fig. 8). We, therefore, predict that Sen1 uses the same binding mechanism to terminate all three yeast RNA polymerases.

Our pre-TC structures could also explain why the termination activity of Sen1 is species specific. Biochemical studies showed that yeast Sen1 and human SETX are unable to cross-terminate human and yeast Pol II, respectively[42]. The mammalian RPB3 domain 2, unlike its yeast ortholog, possesses distinct secondary-structure and surface features. The RPB3 region where SETX putatively binds is not only partially negatively charged but also contains a neutral patch (Extended Data Fig. 7c and Supplementary Fig. 4). Likewise, an AlphaFold model of SETX shows that the region of its β-barrel module (residues 1780–1785: FPADYI) that is comparable to the Pol II-binding region of Sen1 (residues 1187–1192: NRTAVS) does not form an extended β-strand (Extended Data Fig. 9). These differences in structural and surface properties may explain why yeast Sen1 does not bind to mammalian Pol II.

To validate the functional importance of Sen1 binding to Pol II, we generated multiple mutant variants of Sen1. These included domain swaps of the Sen1 β-barrel module from its human functional ortholog SETX (both FL and helicase domain alone) and the yeast nonfunctional homolog Upf1 (helicase domain alone). However, these mutants did not express in *Escherichia coli* or insect cell expression systems (data not shown). Therefore, we studied two variants of Sen1 (R1266E and R1266E_β$_1$-GS) (Extended Data Fig. 10a), impacting the β-barrel module (Fig. 2a,b), and tested their effect on termination in vitro (Methods). Whereas the charge reversal mutant alone (R1266E) had a negligible effect on the termination activity, its combination with the partial disruption of the β$_1$-strand (R1266E_β$_1$-GS) resulted in a ~20% decrease in the termination efficiency of Sen1 (Extended Data Fig. 10b,c). The limited effect of these mutants on the Sen1 termination activity could be because of the intact RNA-binding function of Sen1 along with the highly optimum conditions of the in vitro assay. Complementarily, prior work showed that mutations impacting the Pol II subunits Rpb3 and Rpb11, which disrupt their folding and integrity within the enzyme, cause transcriptional readthrough phenotypes[43]. On this basis, the ability of Sen1 to bind Pol II emerges as a requirement for complete termination, in addition to its ability to bind RNA and its ATPase activity.

Consistent with the need for a specific docking site on Pol II, yeast Sen1 also cannot terminate bacterial RNA polymerase (RNAP)[14]. However, helicase-dependent transcription termination is a prominent mechanism in prokaryotes. Termination of bacterial RNAP requires the helicases Rho[37] and Mfd[44] during transcription and transcription-coupled DNA repair processes, respectively. Rho-dependent transcription termination has been structurally studied, showing how the Rho hexamer directly binds RNAP and its elongation factor NusG to prime RNAP for termination[45]. Single-molecule studies and biochemical characterization revealed that Rho, with its strong helicase activity, terminates RNAP using a combination of the hypertranslocation, allosteric and hybrid-shearing mechanisms[38,46]. Similarly to Sen1 (ref. 14), Rho also requires Pol pausing to cause termination[47].

## Conclusion

In summary, we report the structure of the Sen1-bound Pol II pre-TC in two intermediate states. In combination with biochemical assays, the results lay the foundation for understanding the mechanism of PAS-independent Pol II termination. The structures illustrate that Sen1 binds to Pol II and the nascent RNA transcript in tandem. From there on, Sen1's dynamic ATP-dependent RNA translocase motor can trigger Pol II hypertranslocation, followed by the release of RNA and Pol II from the DNA, leading to termination. While this proposed mechanism is supported by previous data[35,36], future work should address the steps leading to termination in detail to completely rule out other plausible mechanisms[38,39]. During the revision of our paper, independently derived structures of Sen1 FL in free form and its helicase domain alone bound to RNA became available[48]. While the RNA-bound helicase domain structure is essentially similar to our work, the Sen1 FL structure shows how the N-terminal domain (NTD) packs against the helicase domain.

Finally, two unrelated PAS-independent pathways can terminate Pol II transcription in humans. First, the multisubunit integrator complex can terminate Pol II during small nuclear RNA transcription and promoter-proximal transcription attenuation[49,50]. Second, the recently described restrictor complex can terminate Pol II transcription of upstream antisense RNA and short unstable transcripts[5]. However, a prominent role for an enzymatic factor in either of these termination pathways is so far unclear, although XRN2 has been implicated in the integrator pathway[51]. Like Sen1, the INTS7 subunit of the integrator complex also contacts the Pol II RPB3 subunit in humans[52]. It, however, remains to be seen whether these two PAS-independent termination pathways also share other features of Sen1-dependent termination.

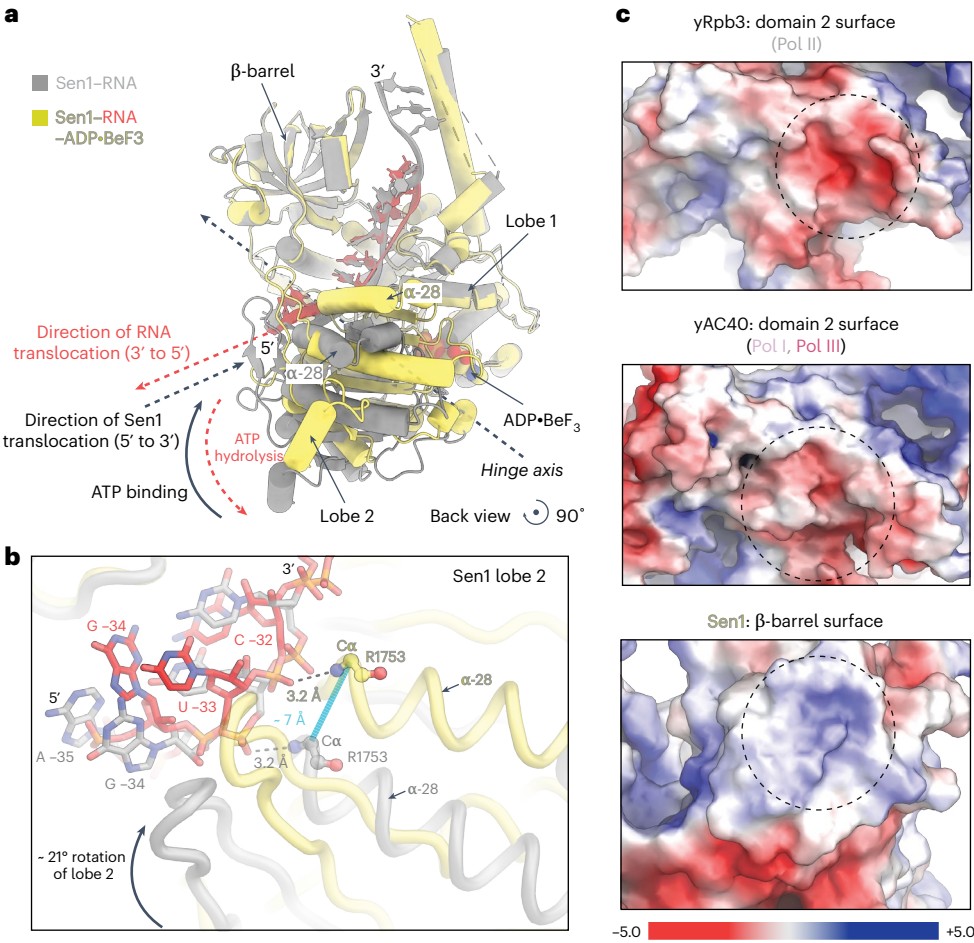

**Fig. 3 | Sen1 ATPase dynamics. a**, Structural comparison of the Pol II pre-TC structure with Sen1 in the presence and absence of ADP·BeF₃. The hinge axis of the Sen1 lobe 2 rotation is shown as a dashed arrow (black). The directions of the lobe 2 movement with respect to ATP binding (solid black) and ATP hydrolysis (dashed red, predicted) are depicted as curved arrows. The directions of Sen1 (black) and RNA (red, predicted) translocation are indicated as dashed arrows. The color codes of individual structures and features are consistent with Extended Data Fig. 6 and labeled accordingly. **b**, Close-up view of Sen1 lobe 2 movement induced by ADP·BeF₃. Both states of Sen1 lobe 2 are represented as ribbons: apo state, gray; ADP·BeF₃-bound state, yellow. The RNA chain corresponding to the

respective states are shown as gray and red sticks, respectively. The distance corresponding to the movement of the Cα atom of residue R1753 in helix α28 of lobe 2 is shown as a dashed line (blue). The hydrogen bonds formed between R1753 and the RNA backbone phosphate in both apo and ADP·BeF₃-bound states are also indicated with dashed lines (black). The curved arrow represents the direction of lobe 2 movement caused by ADP·BeF₃. **c**, Electrostatic potential of the binding surfaces of the yeast Rpb3 (Pol II), AC40 (Pol III) and Sen1 within a range of ±5 $kT/e$. Deep blue, positively charged region; red, negatively charged region (Extended Data Fig. 7). The core interaction patches of all three proteins are highlighted with dashed circles.

## Online content

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

# Methods

## Cloning and protein expression

The FL gene of *Saccharomyces cerevisiae* Sen1 was amplified from complementary DNA by PCR (Addgene, plasmid 99994) and transferred into the 438-C vector (Addgene, plasmid 55220) by ligation-independent cloning[53]. The resulting 438-C Sen1 FL construct contained an N-terminal 6xHis-MBP-tag followed by a modified cleavage site for tobacco etch virus (TEV) protease. Preparation of bacmid and viruses (stage V0 in Sf9 cells (Thermo Fisher, 12659017) and stage V1 in Sf21 cells (Thermo Fisher, 94-003F)) for overexpression of Sen1 FL (hereafter referred to as Sen1) in Hi5 cells (Thermo Fisher, 94-002F) was performed as described previously[54]. Cells expressing Sen1 were harvested by centrifugation (900$g$, 15 min, 4 °C) and resuspended in buffer A (25 mM HEPES pH 7.0, 10% glycerol (v/v), 0.5 mM TCEP, 0.284 µg ml$^{-1}$ leupeptin, 1.37 µg ml$^{-1}$ pepstatin A, 0.17 mg ml$^{-1}$ PMSF and 0.33 mg ml$^{-1}$ benzamidine) containing 500 mM NaCl. The resuspended cells were flash-cooled in liquid nitrogen and stored at −80 °C.

The wild-type Sen1 helicase domain (residues 1095 to 1904) was amplified from the FL gene using PCR and was transferred into the 1-O vector (Addgene, plasmid 29658) by ligation-independent cloning[53]. The construct contained an N-terminal 6xHis-Mocr tag followed by a TEV cleavage site. The mutant R1266E was generated using overlapping primers and the R1266E_β1-GS variant (residues 1187 to 1191 substituted to glycine or serine; N1187S, R1188G, T1189S, A1190G and V1191S) was generated using a round-the-horn cloning strategy. For protein expression, the plasmids were transformed into *E. coli* BL21(DE3)RIL strain (Agilent, 230245). Large-scale cultures were grown from overnight cultures at 37 °C; upon reaching an optical density of 0.6, expression was induced using 1 mM IPTG at 18 °C for 16 h. The cells were then harvested by centrifugation (7,808$g$, 15 min, 4 °C) and resuspended in buffer A containing 500 mM NaCl.

## Protein purification

Recombinant Sen1 was purified by consecutive steps of affinity chromatography, heparin Sepharose chromatography and size-exclusion chromatography. Pellets of the frozen cell suspension were thawed at 25 °C and lysed by sonication. The lysate was clarified by centrifugation (79,000$g$, 60 min, 4 °C), filtered using a 5-µm syringe filter and applied to an amylose resin column. The column was then washed with 15 column volumes of buffer A containing 500 mM NaCl and Sen1 was subsequently eluted using buffer A containing 250 mM NaCl and 100 mM maltose. The eluted protein was incubated overnight with TEV protease. The cleaved 6xHis-MBP-tag was removed from Sen1 using a HiTrap Heparin HP 5-ml column (GE healthcare). After sample application, proteins were eluted over a salt gradient (250 mM to 1 M NaCl) in buffer A for 20 column volumes. The elution fractions were analyzed by SDS–PAGE, of which the Sen1-containing fractions were pooled and concentrated using a VivaSpin concentrator (100-kDa molecular weight cutoff; Sartorius). This sample was then injected onto a Superose6 increase 10/300 GL column (GE healthcare) pre-equilibrated with buffer A containing 300 mM NaCl. Fractions were again analyzed by SDS–PAGE and the homogeneous Sen1 fractions were pooled, concentrated, flash-cooled in liquid nitrogen and stored at −80 °C. Purification of the *S. cerevisiae* Pol II, Spt4, Spt5, Elf1 and Sen1 helicase domain variants were essentially performed as described previously[26,30,55,56]. We also phosphorylated Pol II with CDK7, as the NNS complex has been shown to interact with the phosphorylated (S5) form of the Pol II C-terminal domain (CTD)[57]. To obtain the phosphorylated (S5) form of Pol II for biochemical and structural studies, Pol II was incubated with CDK7 in a 1:25 molar ratio supplemented with 2 mM ATP and purified by size-exclusion chromatography (Superose6 increase 3.2/300 column, GE healthcare). Peak fractions of Pol II were pooled, concentrated and stored at −80 °C.

## Preparation of the pre-TC

We reconstituted the pre-TC complex on a mismatch bubble of the human immunodeficiency virus (HIV) pause scaffold essentially as described previously[58]. Briefly, 220 pmol of preannealed template DNA (5′-GGC AAG CTT TAT TGA GGC TTA AGC AGT GGG TTC AGG GTA CTA GTG TAC ATG CAG ACC G-3′) and RNA (5′-AGU CGU GCG UCU AAU AAC CGG AGA GGG AAC CCA CU-3′) with a 9-bp hybrid and 26 nt of exiting RNA were incubated with 160 pmol of Pol II (phosphorylated, S5) for 10 min at 30 °C. This was followed by adding 300 pmol of the nontemplate (NT) DNA (5′-CGG TCT GCA TGT ACA CTA GTA CCT ACT CGA GTG AGC TTA AGC CTC AAT AAA GCT TGC C-3′) and incubating it for another 10 min. The Spt4–Spt5 complex and Elf1 were then added in fivefold and tenfold molar excess relative to Pol II, respectively, before incubating for 5 min. Next, a twofold excess of Sen1 was added and the final buffer was diluted to correspond to buffer EM (25 mM HEPES pH 7.6, 100 mM KCl, 5 mM MgCl$_2$ and 3 mM TCEP) with 5% glycerol (v/v). This mixture was incubated at 30 °C for 45 min while shaking at 300 r.p.m., followed by centrifugation at 21,000$g$ for 5 min to remove aggregates. The reconstituted pre-TC was subjected to a 10–30% sucrose-gradient centrifugation with simultaneous crosslinking using GraFix[28]. The sample was centrifuged at 175,000$g$ for 16 h at 4 °C. Manual fractions of 200 µl were collected and the crosslinking reaction was quenched using a cocktail of 10 mM aspartate and 30 mM lysine for 10 min. The pre-TC fractions were then dialyzed against buffer EM containing 1% glycerol to remove sucrose and excess glycerol. The pre-TC sample containing ADP·BeF$_3$ was also prepared using the same reconstitution scheme except that 1 mM ADP·BeF$_3$ was supplemented during the incubation of the sample before the sucrose gradient.

## Cryo-EM grid preparation, data collection and processing

Quantifoil R3.5/1 holey carbon grids were precoated with a homemade amorphous continuous carbon (3 nm) and glow-discharged for 45 s. Then, 4 µl of pre-TC ± ADP·BeF$_3$ was incubated with the grids for 3 min inside the Vitrobot chamber equilibrated at 4 °C, 100% humidity. The grid was blotted for 3 s and vitrified by plunge-freezing in liquid ethane. Single-particle cryo-EM data were collected on a 300-kV FEI Titan Krios (Thermo Fisher) with a K3 summit direct detector (Gatan) and a GIF quantum energy filter (Gatan) operated with a slit width of 20 eV. Data collection was automated with SerialEM[59] at a nominal magnification of ×81,000, corresponding to a pixel size of 1.05 Å per pixel. For the pre-TC sample, 48,735 image stacks, with each stack containing 40 frames, were collected at a defocus range of −0.5 to −3.0 µm. All video frames were contrast transfer function (CTF)-estimated, motion-corrected and dose-weighted using Warp[60]. Particles were picked by Warp using a locally trained neural network, resulting in 9,710,589 particles as a starting set. Subsequent steps of image processing were performed with cryoSPARC (version 3.2.0)[61] and RELION (version 3.1.0)[62].

Exported particles from Warp were extracted using RELION (version 3.1.0) with a binning factor of 2 and a box size of 180 pixels (pixel size of 2.1 Å per pixel) to perform initial clean-up and sorting. To achieve this, iterative rounds of two-dimensional (2D) classification, followed by heterogeneous and homogeneous refinements were performed in cryoSPARC. This strategy helped identify the subset of 630,478 particles containing all the components of pre-TC. This set was then re-extracted without binning and processed using RELION (version 3.1.0). The particles were subjected to 3D refinement, followed by two rounds of CTF refinement before and after Bayesian polishing. The particles were then sorted for Sen1 occupancy through focused 3D classification with a large spherical mask (mask 1) encompassing the region of Sen1. The resulting 95,644 particles were then 3D-refined with and without mask 1 to obtain the consensus map (map 1) of Sen1-containing pre-TC at 2.8 Å. To improve the quality of the Sen1 region, multibody refinement was performed using mask 1 for Sen1 and another spherical mask (mask 2) encompassing the remainder of the pre-TC. This yielded a local map of Sen1 extending to 3.3-Å resolution (map 2).

For the pre-TC ADP·BeF$_3$ sample, 29,640 image stacks, with each stack containing 40 frames, were collected at a defocus range of −0.3 to −2.5 μm. Preprocessing of the video frames and autopicking of particles were performed using Warp[60] as explained above for the pre-TC. A total of 4,333,083 particles exported from Warp were extracted in RELION (version 3.1.0)[62] with a binning factor of 2 and a box size of 180 pixels (a pixel size of 2.1 Å per pixel). Initial clean-up and sorting were performed as explained for pre-TC using cryoSPARC (version 3.2.0)[61], resulting in 546,774 particles. While we observed two different particle populations of pre-TC with varying occupancies for parts of Spt4 and Spt5, the best reconstruction for the Sen1 region was obtained in the particle class with a poor occupancy for these proteins in the upstream DNA region. This behavior of Spt4 and Spt5 is dependent on ice thickness and is previously known. Both sets of particles with (set 1) and without (set 2) the parts of Spt4 and Spt5 from cryoSPARC were then re-extracted without binning and processed using RELION (version 3.1.0). After performing the CTF refinement and Bayesian polishing cycle followed by focused 3D classification using mask 1, we obtained a final set of 9,095 particles from set 1. Next, 3D refinement of this particle set yielded a consensus map (map 3) of the pre-TC ADP·BeF$_3$ at 4.3-Å resolution (map 3). Set 2 imported from cryoSPARC was also sorted further with the process used for set 1, resulting in a final set of 15,633 particles after focused 3D classification using mask 1. Multibody refinement of this particle set using masks 1 and 2 yielded a Sen1 local map extending to 4.4-Å resolution (map 4).

The resolutions of the final maps (1–4) were calculated using the gold-standard Fourier shell correlation (FSC) 0.143 criterion. Local resolution estimates were obtained using the RELION inbuilt local resolution tool with a $B$ factor of 0. The local resolution filtered maps obtained from this tool were then locally sharpened using PHENIX. auto_sharpen[63] to assist model building.

## Model building and refinement

The Pol II, nucleic acid strands (DNA and RNA), Spt4, Spt5 (KOW 5) and Sen1 subunits of both pre-TC structures were modeled using previously published structures (Protein Data Bank (PDB) 7NKX and 6I59). Subunits Spt5 (NGN, KOW 1L, 2, 3 and x4 domains) and Elf1 were built using AlphaFold[29] models. Although we phosphorylated the Pol II CTD using CDK7, we did not observe any density representing Pol II CTD or Sen1 NTD in our cryo-EM maps. The initial models of all the subunits were rigid-body fitted into the density using UCSF Chimera[64] and were manually extended and corrected using Coot[65] to fit the density. Map regions with ambiguous density corresponding to linkers were not modeled. The sequence registers of DNA and RNA strands were manually mutated to fit the experimental data for the apo structure. For the structure of Sen1 bound to ADP·BeF$_3$, the RNA register was retained as observed in the apo structure and the RNA trajectory was remodeled to avoid atomic clashes. The models were then subjected to iterative rounds of PHENIX real-space refinement[63] followed by manual adjustment in Coot to achieve the final models (Extended Data Fig. 5). The stereochemistry assessment of the final models was performed using MolProbity[66]. Details of the final model confidence are furnished in Table 2. Representations of 3D structures and maps in the figures were prepared using PyMOL, UCSF Chimera and UCSF ChimeraX.

## In vitro transcription termination assay

Termination assays were performed basically as previously described (Extended Data Fig. 1)[67]. For this assay, we used a modified HIV pause scaffold (template strand: 5′-TGC CGT GAC TTG GCA ACG TCG GTC TGC ATG TAC ACT AGT ACC TGG AAC CCA CTA TCA ACT ATA ATC CTC AAC CAT AAG GGG GGA ATC CGC ATC ATG ATG C; NT strand: 5′-GCA TCA TGA TGC GGA TTC CCC CCT TAT GGT TGA GGA TTA TAG TTG ATA GTG GGT TCC AGG TAC TAG TGT ACA TGC AGA CCG ACG TTG CCA AGT) in the cryo-EM experiment without the mismatch bubble and a longer downstream DNA region amenable for Pol II transcription. The 5′ end

of the NT strand was labeled with a biotin tag. A short FAM-labeled 11-nt RNA primer (5′-GGGAACCCACU-3′) was used to promote transcription. Pol II (phosphorylated, S5) together with the elongation factors Spt4, Spt5 and Elf1 were assembled on the HIV pause scaffold using the scheme explained above for cryo-EM sample preparation. The assembled Pol II EC was then immobilized on streptavidin beads (Dynabeads MyOne Streptavidin T1 from Invitrogen) and washed using assay buffer (25 mM HEPES pH 7.5, 100 mM NaCl, 8 mM MgCl$_2$, 10 μM ZnCl$_2$, 10% glycerol and 3 mM TCEP) containing 0.1% Triton X-100 followed by 500 mM NaCl to remove the Pol II EC molecules nonspecifically bound to the beads. Transcription was then initiated using a mixture of ATP, cytidine triphosphate and uridine triphosphate (1 mM each) with and without Sen1 (50 nM) and stopped after 10 min using 0.5 M EDTA. The beads and the supernatant fractions were then separated followed by resuspension of the bead fraction in 8 μl of loading buffer (1× Tris–borate–EDTA, 8 M urea). The RNA in the supernatant fraction was enriched as described previously[68] and resuspended in 8 μl of loading buffer. Then, 2 μl each of the bead and supernatant fractions were separated on 15% urea–PAGE and analyzed with a Typhoon 9500 FLA imager (GE Healthcare Life Sciences).

The assays performed with Sen1 helicase domain variants were conducted for 5 min, then stopped using 0.5 M EDTA and analyzed as explained above. Gels then underwent linear contrast enhancement. The data for all quantified termination assays in Fig. 1 are available in the Source data. Each termination assay was conducted independently and repeated three times. The quantification of RNA products was carried out using Fiji 1.0. These products were normalized against the background intensity and the termination efficiency was calculated from the product of RNA amount in the supernatant fraction divided by the total RNA in the beads and supernatant fraction of the respective sample. The dot plot was generated using ggplot2 and shows the values from each experimental replicate along with the mean values. Statistical significance was determined using a paired two-tailed $t$-test, with *$P < 0.05$, **$P < 0.01$ and ***$P < 0.001$ considered significant.

## Statistics and reproducibility

No statistical method was used to predetermine sample size. No data were excluded from the analyses. The experiments were not randomized. The investigators were not blinded to allocation during experiments and outcome assessment.

## Reporting summary

Further information on research design is available in the Nature Portfolio Reporting Summary linked to this article.

## Data availability

The cryo-EM density reconstructions and models were deposited to the EM Data Bank (Pol II pre-TC overall map, EMD-19019; Sen1–RNA apo local map, EMD-19020; Pol II pre-TC ADP·BeF$_3$ overall map, EMD-19022; Sen1–RNA ADP·BeF$_3$ local map, EMD-19021) and their respective coordinate files were deposited to the PDB (Pol II pre-TC overall structure, 8RAM; Sen1–RNA apo structure, 8RAN; Pol II pre-TC ADP·BeF$_3$ overall structure, 8RAP; Sen1–RNA ADP·BeF$_3$ structure, 8RAO). All data are available in the main text or Supplementary Information. Source data are provided with this paper.

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

## Acknowledgements

We thank members of the Cramer laboratory for help and discussions, particularly C. Dienemann and M. Engeholm for critically reading the paper. We also thank J. Schmitzova and M. Ochmann for discussions. We thank F. Grabbe for purifying the yeast Pol II and CDK7 proteins. We thank C. Dienemann and U. Steuerwald for maintenance of the EM facility. We thank C. Dienemann for help during cryo-EM data collection. S.R. was supported by a Peter and Traudl Engelhorn Foundation postdoctoral fellowship. P.C. was supported by the Deutsche Forschungsgemeinschaft (EXC 2067/1-390729940, SFB860).

## Author contributions

S.R. conceptualized the project and carried out all experiments, unless stated otherwise. S.R., T.H. and C.O. performed the purification of Sen1 mutants. S.R. performed data visualization and analysis with assistance from M.L. S.R. and P.C. interpreted the data and wrote the paper with input from M.L.

## Funding

## Competing interests

The authors declare no competing interests.

## Additional information

**Extended data** is available for this paper at https://doi.org/10.1038/s41594-024-01409-0.

**Correspondence and requests for materials** should be addressed to Srinivasan Rengachari or Patrick Cramer.

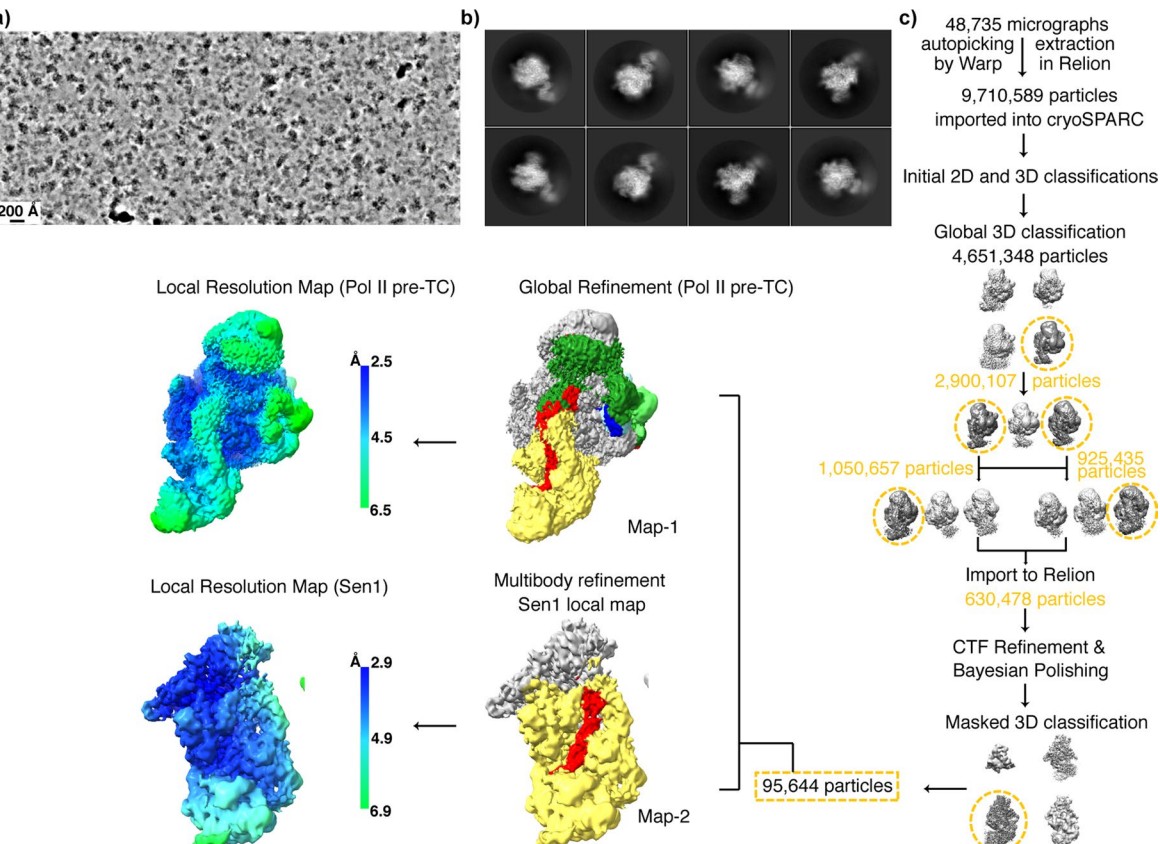

**Extended Data Fig. 1 | Cryo-EM data processing of the Pol II pre-TC.**
**a**. Representative cryo-EM micrograph (replicated more than 45,000 times) showing the particle distribution per field of view in the pre-TC dataset (denoised in Warp). Scale bar – 200 Å. **b**. Representative 2D class averages showing different orientations of the pre-TC. **c**. Complete processing scheme of the pre-TC.

After initial clean-up followed by successive rounds of 3D classification sorting, a final set of 95,644 particles was obtained. The final maps are coloured using the subunit colour code in Fig. 1. The local resolution maps indicate the resolution range of the final maps (scale bar).

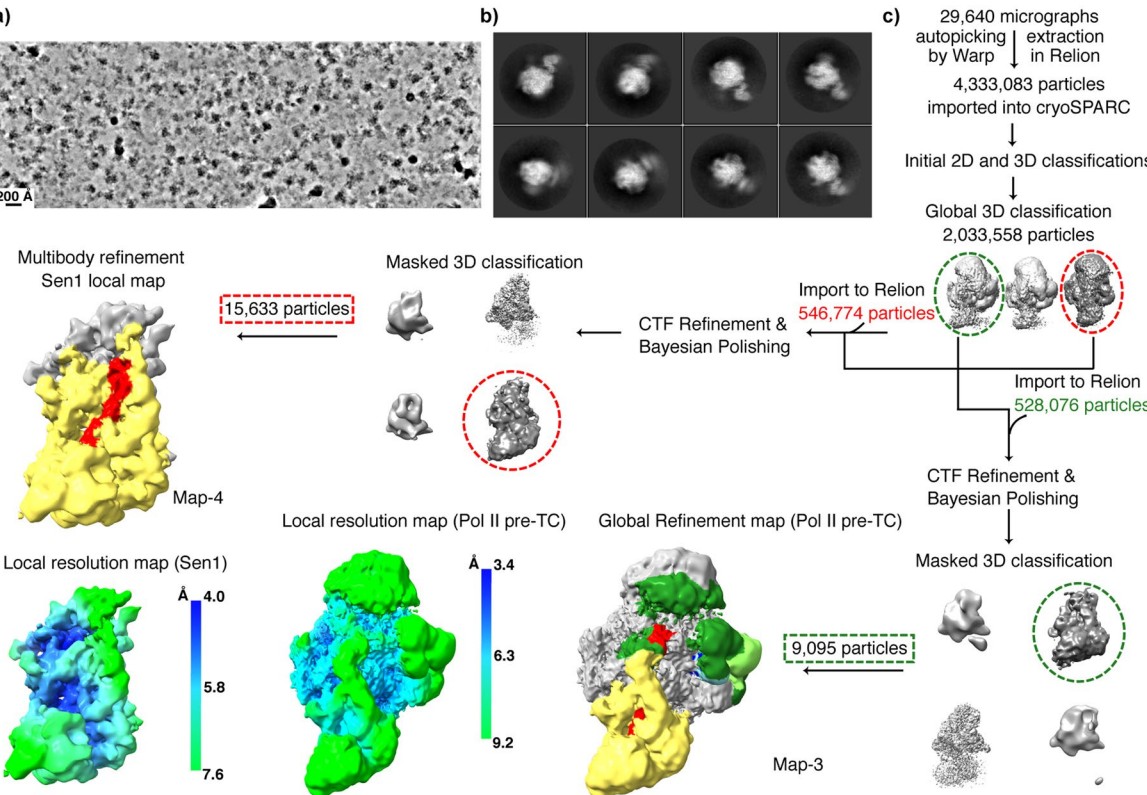

**Extended Data Fig. 2 | Cryo-EM data processing of the Pol II pre-TC bound to ADP·BeF3. a**. Representative cryo-EM micrograph (replicated more than 20,000 times) showing the particle distribution per field of view in the pre-TC dataset (denoised in Warp). Scale bar – 200 Å. **b**. Representative 2D class averages showing different orientations of the pre-TC ADP·BeF3 complex. **c**. Complete processing scheme of the pre-TC. After initial clean-up followed by successive rounds of 3D classification sorting, a final set of 15,633 particles was obtained. The final maps are coloured using the subunit colour code in Fig. 1. The local resolution maps indicate the resolution range of the final maps (scale bar).

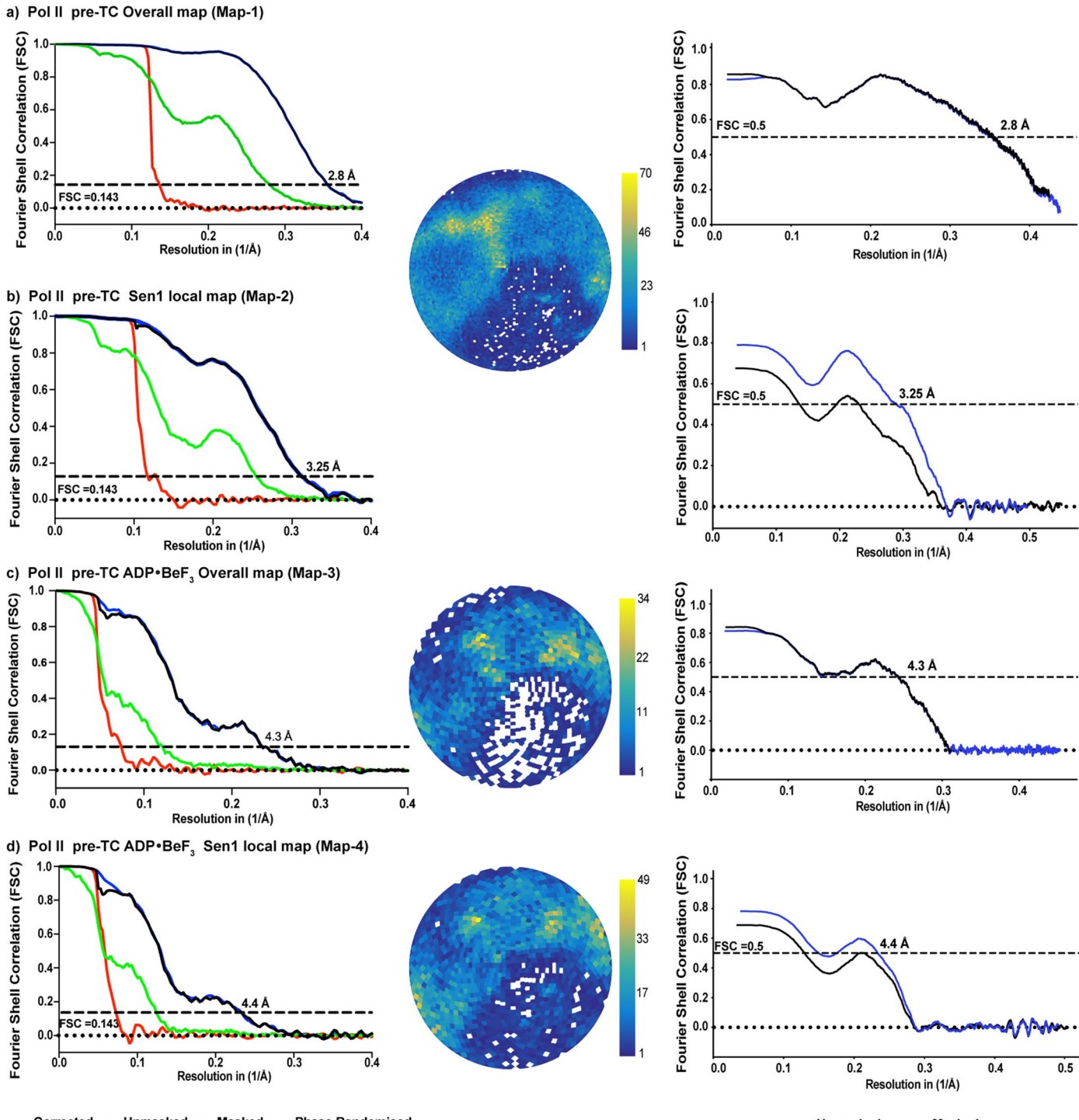

a) Pol II pre-TC Overall map (Map-1)

b) Pol II pre-TC Sen1 local map (Map-2)

c) Pol II pre-TC ADP•BeF₃ Overall map (Map-3)

d) Pol II pre-TC ADP•BeF₃ Sen1 local map (Map-4)

—— Corrected —— Unmasked —— Masked —— Phase Randomised

—— Unmasked —— Masked

**Extended Data Fig. 3 | Resolution plots of cryo-EM reconstructions.**
**a–d.** On the left - FSC plot showing the overall resolution of the reconstructions determined by the gold standard FSC cut-off 0.143, indicated in the graph. In the middle – angular distribution plot of the respective reconstruction showing assignment of particles with respect to various angles. Colour bar indicates number of particles per angular bin (white areas indicate unpopulated angles). On the right - Model-to-map FSCs, showing the fit of modelled structures to their corresponding maps.

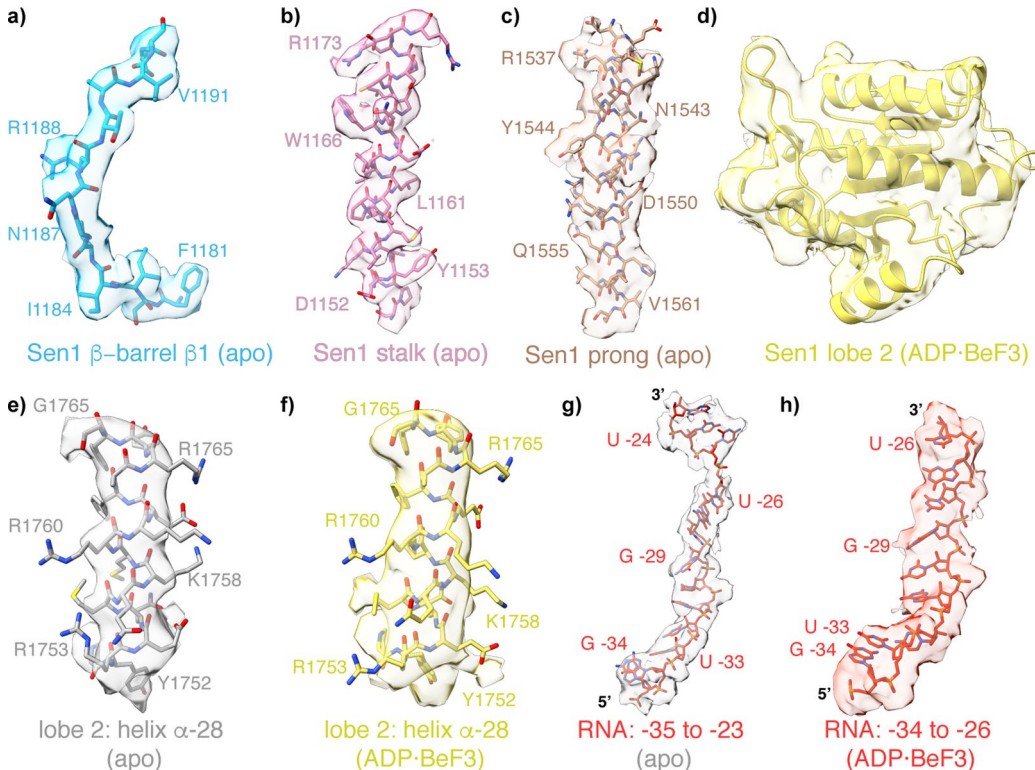

**Extended Data Fig. 4 | Map quality and map to model fit. a–h.** Regions of cryo-EM maps of pre-TC structures in apo and ADP·BeF3 bound states. The maps are displayed as iso-surfaces with colours matching the models represented as sticks. The corresponding regions of the model are labelled accordingly.

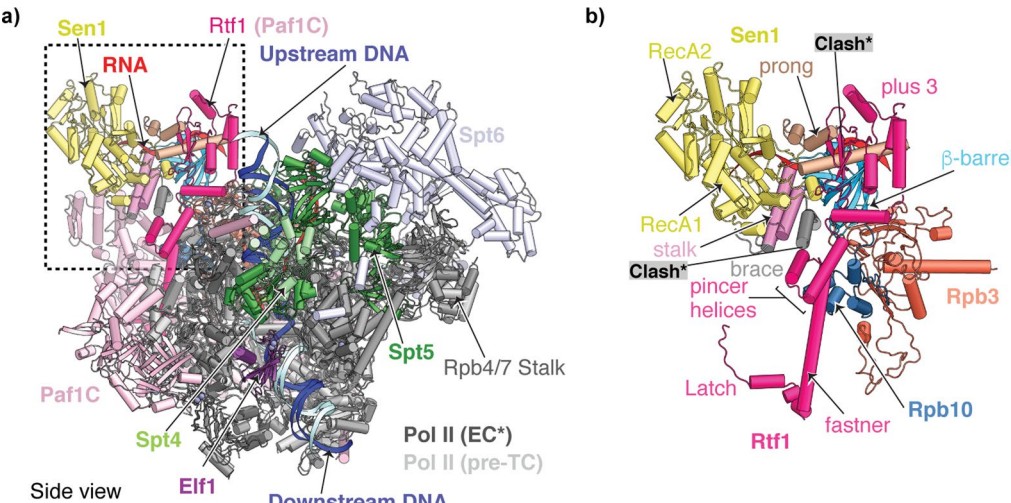

**Extended Data Fig. 5 | Comparison of Pol II pre-TC and Pol II EC*. a)** Structural superposition of Pol II pre-TC to Pol II EC* bound to the elongation factors Spt4, Spt5, Spt6 and Elf1. The region corresponding to the putative clash between Sen1 and Rtf1 is highlighted as a dashed box. **b)** Panel **b** shows a close-up of the proteins and chains involved in the clash in panel **a**.

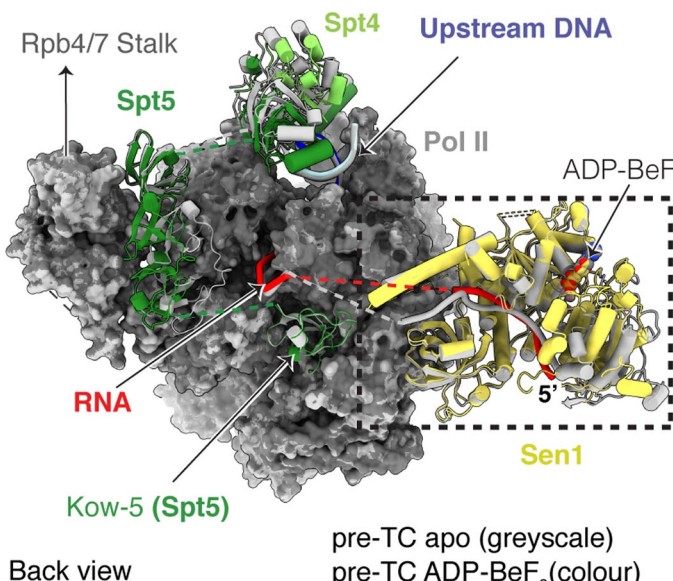

**Extended Data Fig. 6 | Comparison of apo and ADP·BeF3 bound pre-TC structures.** Overall view of pre-TC structures in apo and ADP·BeF3 bound states, superposed to each other. The region corresponding to RNA bound Sen1 presenting the movement of ATPase lobe2 is highlighted with a dashed box. ADP·BeF3 is shown as spheres. The representation and colour codes of the pre-TC components are consistent with Fig. 1.

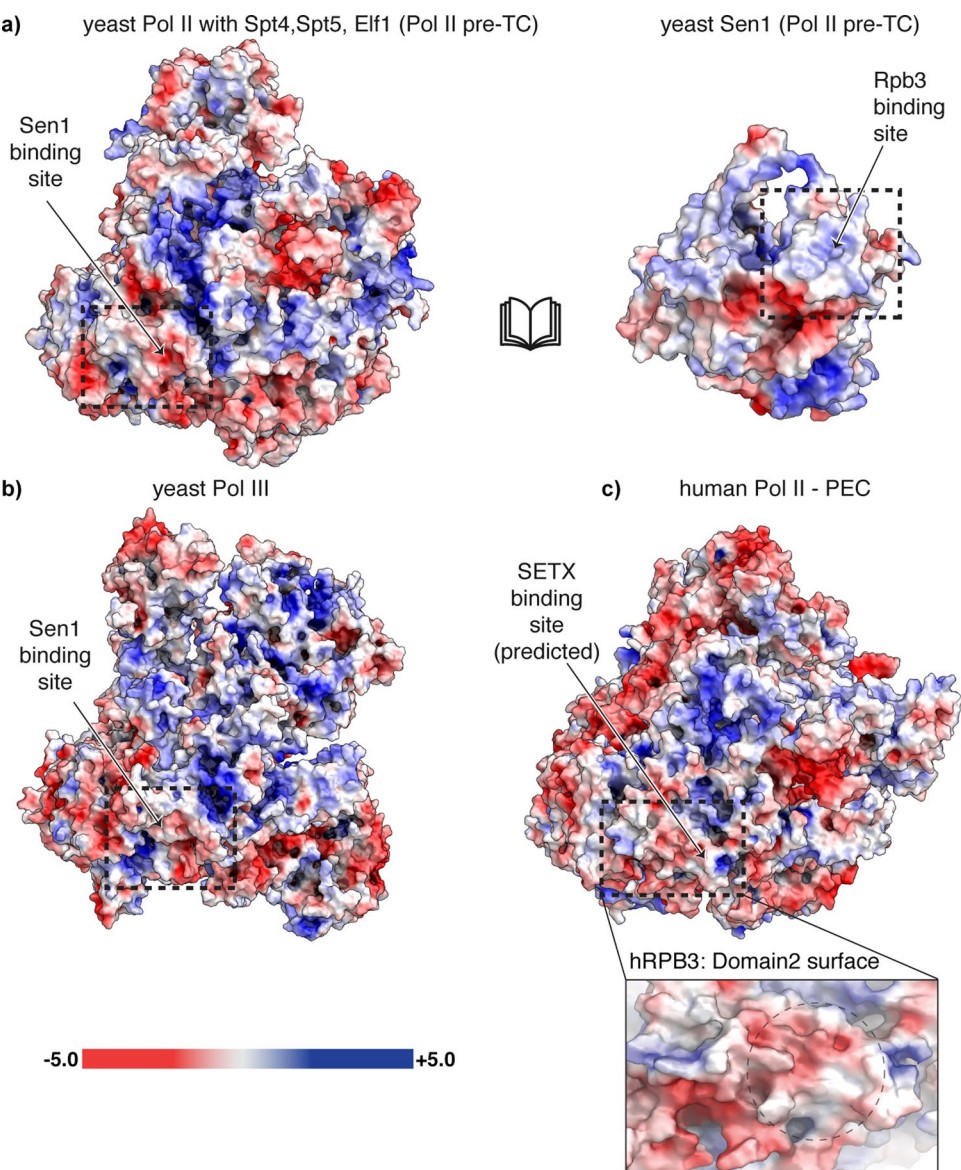

**Extended Data Fig. 7 | Electrostatic surface potential analysis. a–c**. The electrostatic surface potential of yeast Pol II and Sen1 (**a**), yeast Pol III (**b**) and human Pol II (**c**) with a range of ± 5 kT/e, where deep blue represents positively and red represents negatively charged areas. Panel **a** corresponds to a book view of the pre-TC. The dashed boxes represent the binding surfaces involved in complex formation (Fig. 3c). The interaction surface of the human Pol II RPB3 subunit is shown as an inset.

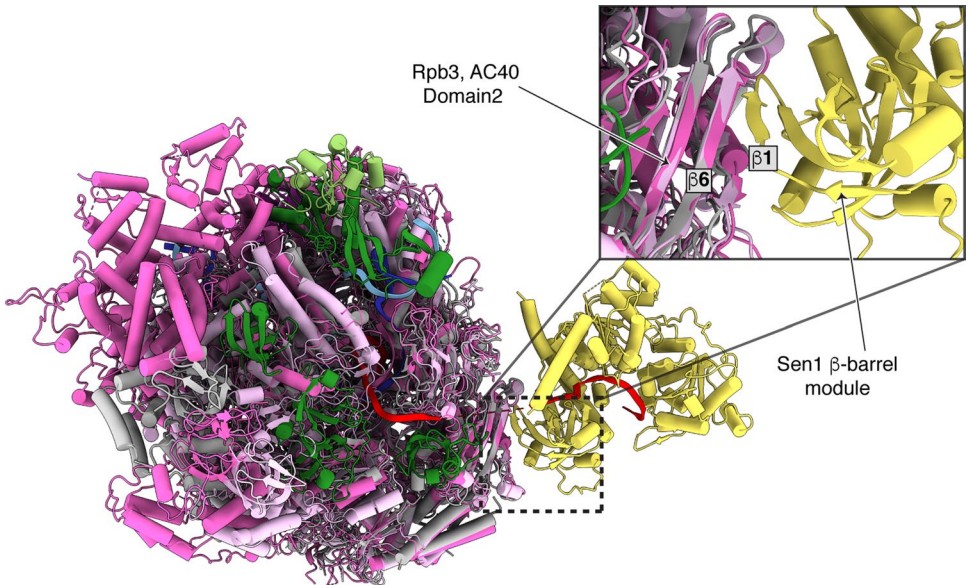

**Extended Data Fig. 8 | Comparison of yeast Pol II pre-TC with Pol I and Pol III.** Structural superposition of Pol I (PDB code: 4C3M), Pol II pre-TC (this work) and Pol III (PDB code: 5FJ8) to compare the conservation of the Sen1 binding interface between Rpb3 and AC40. Inset shows the close up view of the superposed AC40 amenable for binding Sen1 similar to Rpb3. The components are labelled with colour codes.

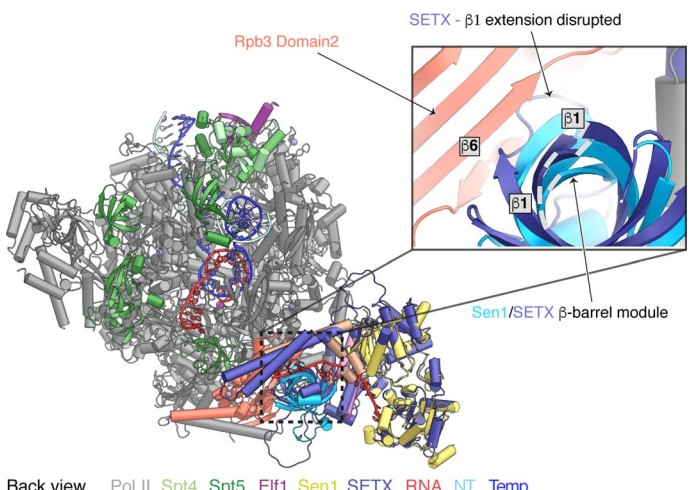

Back view  Pol II  Spt4  Spt5  Elf1  Sen1  SETX  RNA  NT  Temp

**Extended Data Fig. 9 | Comparison of yeast Sen1 and human SETX ß-barrel modules.** Structural superposition of human SETX helicase domain (AlphaFold2 model) to Pol II pre-TC (this work). The inset shows the comparison of the secondary structural differences between Sen1 and SETX in the ß1-strand of their respective ß-barrel module.

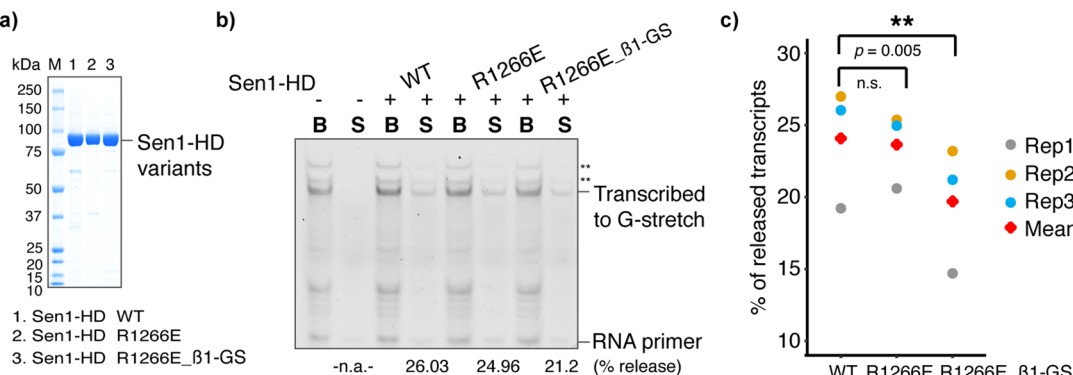

**Extended Data Fig. 10 | Functional validation of Sen1-Pol II interaction surface. a**. SDS-PAGE analysis (replicated minimum thrice) showing purified variants of Sen1 helicase domain (Sen1-HD). **b**. Denaturing gel showing the termination activity of the purified Sen1 variants from panel **a**. The final products corresponding to terminated transcripts are marked and labelled. B – beads fraction; S – supernatant fraction. Asterisks indicate the longer transcripts resulting from mis-incorporation. The experiment was repeated minimum thrice. **c**. Dot plot showing the percentage of released, that is terminated, transcripts using the Sen1 variants from panel **a**, quantified (see Methods) for three replicates of termination assays (panel **b** and Source Data Fig. 1). The effect of Sen1 mutation was tested by paired two-tailed t-test.

# nature research

# Reporting Summary

Nature Research wishes to improve the reproducibility of the work that we publish. This form provides structure for consistency and transparency in reporting. For further information on Nature Research policies, see our Editorial Policies and the Editorial Policy Checklist.

## Statistics

For all statistical analyses, confirm that the following items are present in the figure legend, table legend, main text, or Methods section.

| n/a | Confirmed | |
|---|---|---|
| ☐ | ☒ | The exact sample size (*n*) for each experimental group/condition, given as a discrete number and unit of measurement |
| ☒ | ☐ | A statement on whether measurements were taken from distinct samples or whether the same sample was measured repeatedly |
| ☐ | ☒ | The statistical test(s) used AND whether they are one- or two-sided<br>*Only common tests should be described solely by name; describe more complex techniques in the Methods section.* |
| ☒ | ☐ | A description of all covariates tested |
| ☒ | ☐ | A description of any assumptions or corrections, such as tests of normality and adjustment for multiple comparisons |
| ☐ | ☒ | A full description of the statistical parameters including central tendency (e.g. means) or other basic estimates (e.g. regression coefficient) AND variation (e.g. standard deviation) or associated estimates of uncertainty (e.g. confidence intervals) |
| ☒ | ☐ | For null hypothesis testing, the test statistic (e.g. *F*, *t*, *r*) with confidence intervals, effect sizes, degrees of freedom and *P* value noted<br>*Give P values as exact values whenever suitable.* |
| ☒ | ☐ | For Bayesian analysis, information on the choice of priors and Markov chain Monte Carlo settings |
| ☒ | ☐ | For hierarchical and complex designs, identification of the appropriate level for tests and full reporting of outcomes |
| ☒ | ☐ | Estimates of effect sizes (e.g. Cohen's *d*, Pearson's *r*), indicating how they were calculated |

*Our web collection on statistics for biologists contains articles on many of the points above.*

## Software and code

Policy information about availability of computer code

| Data collection | Serial EM 3.8 beta 8 |
|---|---|
| Data analysis | RELION 3.0.17, UCSF Chimera 1.13, UCSF ChimeraX v1.11, Pymol 2.3.4, Coot 0.8.9.2, Warp v1.0.7-1.0.9, PHENIX 1.18.2, cryoSPARC v3.2.0, T-Coffee Sequence alignment algorithm |

For manuscripts utilizing custom algorithms or software that are central to the research but not yet described in published literature, software must be made available to editors and reviewers. We strongly encourage code deposition in a community repository (e.g. GitHub). See the Nature Research guidelines for submitting code & software for further information.

## Data

Policy information about availability of data

All manuscripts must include a data availability statement. This statement should provide the following information, where applicable:
- Accession codes, unique identifiers, or web links for publicly available datasets
- A list of figures that have associated raw data
- A description of any restrictions on data availability

The cryo-EM density reconstructions and models are deposited to the EMDB (Pol II pre-TC overall map – 19019; Sen1-RNA apo local map – 19020; Pol II pre-TC ADP · BeF3 overall map – 19022; Sen1-RNA ADP · BeF3 local map – 19021) and their respective coordinate files will be deposited to the PDB (Pol II pre-TC overall structure – 8RAM; Sen1-RNA apo structure – 8RAN; Pol II pre-TC ADP · BeF3 overall structure – 8RAP; Sen1-RNA ADP · BeF3 structure – 8RAO). The PDB models used as inputs in this study - 2XZO, 6I59, 7NKX. All data are available in the main text, tables or the supplementary materials.

# Field-specific reporting

Please select the one below that is the best fit for your research. If you are not sure, read the appropriate sections before making your selection.

☒ Life sciences ☐ Behavioural & social sciences ☐ Ecological, evolutionary & environmental sciences

For a reference copy of the document with all sections, see nature.com/documents/nr-reporting-summary-flat.pdf

# Life sciences study design

All studies must disclose on these points even when the disclosure is negative.

| | |
|---|---|
| Sample size | No Sample size calculations were performed. Experiments were performed with three sample replicates and each experiment was repeated minimum thrice. This is a standard in the field and the sample size was sufficient to observe the effect and binary outcome of these experiments. i.e. Pol II in vitro transcription termination assay |
| Data exclusions | No data were excluded from the analyses. |
| Replication | All attempts at replication were successful. Cryo-EM single particle analysis inherently relies on averaging over a large number of independent observations. For the Pol II pre-TC data - 48,735 independent micrographs were collected and for Pol II pre-TC ADP.BeF3 data - 24,690 independent micrographs were collected. |
| Randomization | Samples were not allocated to groups. |
| Blinding | Investigators were not blinded during data acquisition and analysis because it is not a common procedure for the methods employed. |

# Reporting for specific materials, systems and methods

We require information from authors about some types of materials, experimental systems and methods used in many studies. Here, indicate whether each material, system or method listed is relevant to your study. If you are not sure if a list item applies to your research, read the appropriate section before selecting a response.

### Materials & experimental systems

| n/a | Involved in the study |
|---|---|
| ☒ | ☐ Antibodies |
| ☐ | ☒ Eukaryotic cell lines |
| ☒ | ☐ Palaeontology and archaeology |
| ☒ | ☐ Animals and other organisms |
| ☒ | ☐ Human research participants |
| ☒ | ☐ Clinical data |
| ☒ | ☐ Dual use research of concern |

### Methods

| n/a | Involved in the study |
|---|---|
| ☒ | ☐ ChIP-seq |
| ☒ | ☐ Flow cytometry |
| ☒ | ☐ MRI-based neuroimaging |

## Eukaryotic cell lines

Policy information about cell lines

| | |
|---|---|
| Cell line source(s) | Hi5 cells: Expression Systems, Tni Insect cells in ESF921 media, item 94-002F<br>Sf9 cells: ThermoFisher, Catalogue Number 12659017, Sf9 cells in Sf-9000TM III SFM Sf21 cells: Expression Systems, SF21 insect cells in ESF921 medium, Item 94-003F. E.coli BL21(DE3)RIL strain, Agilent, Catalogue number 230245. |
| Authentication | None of the cell lines were authenticated. |
| Mycoplasma contamination | Cell lines were not tested for mycoplasma contamination. |
| Commonly misidentified lines<br>(See ICLAC register) | No commonly misidentified cell lines were used. |

