## [Peer Review File · Nature Structural & Molecular Biology]

Mechanism of polyadenylation independent RNA polymerase II termination

Corresponding Author: Dr Srinivasan Rengachari

This manuscript has been previously reviewed at another journal. This document only contains reviewer comments, rebuttal and decision letters for versions considered at Nature Structural & Molecular Biology.

Version 0:

Decision Letter:

Our ref: NSMB-A49150-T

1st May 2024

Dear Dr. Rengachari,

Thank you for submitting your revised manuscript "Mechanism of polyadenylation independent RNA polymerase II termination" (NSMB-A49150-T). We'll be happy in principle to publish it in Nature Structural & Molecular Biology, pending minor revisions to satisfy the referees' final requests and to comply with our editorial and formatting guidelines. The final revisions should focus on textually satisfying the lingering concerns of the reviewers with respect to the demonstrated in vivo significance of the findings, discussing the limitations of the current mutational analyses, and its associated results, and toning down claims where needed.

To facilitate our work at this stage, it is important that we have a copy of the main text as a word file. If you could please send along a word version of this file as soon as possible, we would greatly appreciate it; please make sure to copy the NSMB account (cc'ed above).

Sincerely,

Katarzyna Ciazynska, PhD
(she/her)
Associate Editor
Nature Structural & Molecular Biology
<https://orcid.org/0000-0002-9899-2428>

Version 1:

Decision Letter:

25th Sep 2024

Dear Dr. Rengachari,

We are now happy to accept your revised paper "Mechanism of polyadenylation independent RNA polymerase II termination" for publication as an Article in Nature Structural & Molecular Biology.

Your paper will be published online soon after we receive proof corrections and will appear in print in the next available issue. You can find out your date of online publication by contacting the production team shortly after sending your proof corrections.

Please note that *Nature Structural & Molecular Biology* is a Transformative Journal (TJ). Authors may publish their research with us through the traditional subscription access route or make their paper immediately open access through payment of an article-processing charge (APC). Authors will not be required to make a final decision about access to their article until it has been accepted. [Find out more about Transformative Journals](https://www.springernature.com/gp/open-research/transformative-journals)

Authors may need to take specific actions to achieve [compliance](https://www.springernature.com/gp/open-research/funding/policy-compliance-faqs) with funder and institutional open access mandates. If your research is supported by a funder that requires immediate open access (e.g. according to a

[Plan S principles](https://www.springernature.com/gp/open-research/plan-s-compliance)) then you should select the gold OA route, and we will direct you to the compliant route where possible. For authors selecting the subscription publication route, the journal's standard licensing terms will need to be accepted, including [self-archiving policies](https://www.springernature.com/gp/open-research/policies/journal-policies). Those licensing terms will supersede any other terms that the author or any third party may assert apply to any version of the manuscript.

Sincerely,

Dimitris Typas
Senior Editor
Nature Structural & Molecular Biology
ORCID: 0000-0002-8737-1319

Responses are in *blue italics*

Referee #1 (Remarks to the Author):

In the revised version of the manuscript the authors have made considerable efforts to improve the text and have introduced many of the modifications requested. The message of the revised version is clearly more accurate and generally improved. However, the most important conclusion of this work is still not sufficiently supported by the data to meet the high-quality requirements of a Nature paper.

We thank the reviewer for the assessment and we have addressed the comments as described below.

The work focuses on the interaction of Sen1 β 1-strand with the β 6-strand of Rpb3, and the model posits that this interaction is crucial for termination. Because this interaction is not altered in the transition state analogue (i.e. the ADP-BeF3-bound form of the Sen1-Pol II complex) while the RNA is translocated by 1 nucleotide relative to the apo-form, the authors propose that upon ATP hydrolysis the RNA is pulled away from the Pol II catalytic center, thus inducing termination. The interaction between Rpb3 and Sen1 is also central to the speculation for why Sen1 can also terminate Pol III and Pol I transcription and why human Senataxin cannot terminate yeast Pol II.

To prove this model based on structural data, the authors have introduced 6 mutations in Sen1 that are expected to affect the interaction with Rpb3 (charge inversion in R1266E coupled to mutation of aa. 1187-1191 to G or S). According to the model, these mutations were expected to disrupt the Rpb3-Sen1 interaction significantly. and induce a major termination defect, but only a minor defect was observed (a decrease of 20%). This result is hardly compatible with the notion that the Sen1-Rpb3 interaction is central to the mechanism of termination, especially considering that the mutations introduced might generally affect the function of Sen1 (which could also be verified, at least for the helicase and ATPase activities). It is certainly possible that the mutations introduced only partially affect the Rpb3-Sen1 interaction (which could also be assessed), but even in this case the model remains unproven.

Therefore, I still cannot recommend publication in Nature. Nevertheless, the results are overall interesting and I believe this study could be redirected to a more specialized, high visibility journal. Even in this case, however, my suggestion to the authors would be to present the model in a more cautious manner, as alternative or complementary mechanistic options are not excluded by their data.

We thank the reviewer for the suggestions. As presented in our previous response, the attempts to generate a mutant of Sen1 with a stronger termination defect did not work due to protein expression issues. So, we would like to emphasize again that the challenge is likely technical rather than experimental. And as alluded by the reviewer, we had also cautiously rephrased the statements on the mechanism of termination to ensure that alternate possibilities are not excluded.

Referee #2 (Remarks to the Author):

This reviewer is satisfied with the author's revision

We thank the reviewer for the positive assessment of our manuscript.

Referee #3 (Remarks to the Author):

The authors have addressed most of my previous comments; however, there are two minor points they should still address before publication.

We thank the reviewer for the comments and we have addressed them as described below.

1. I appreciate the authors' attempt to prepare Sen1 mutant variants to confirm the importance of the interaction between Sen1 and Rpb3 in termination. However, the mutants the authors created had a very weak effect on the termination activity. The authors should conduct a binding assay to test the affinity between these Sen1 mutant variants and Pol II, and combine this assay with the termination assay to analyze the results.

We thank the reviewer for the suggestion. Though binding assays with mutants are commonly employed to test the importance of protein-protein interfaces, it is known that Sen1 binding to the Pol II machinery requires the nascent RNA (PMID: 23748379). A lack of stronger effect of the mutations on termination activity is likely due to technical challenges. Kindly see above. (Response to Reviewer 1)

More importantly, in vivo data are needed to enrich the story and make it more convincing.

We thank the reviewer for this suggestion. While we agree with the general importance of performing in vivo validation experiments, this is most likely not feasible in a timely manner. There are many factors to consider, including the large size of the Sen1 gene (~ 8kb) and the position of the beta-barrel module in the middle of the gene. Therefore, this experiment is beyond the scope of this manuscript.

2. The most recent papers on pre-termination show that both Integrator and Rat1-Rai1 compete with DSIF on Pol II, which is important for termination. Therefore, my concern is that DSIF might not engage with Sen1-bound Pol II in the in vivo context.

We thank the reviewer for this comment. We acknowledge the recent works on Rat1-Rai1 and Integrator, showing the interplay between these complexes and DSIF. However, our yeast pre-TC structures and their comparison with the Pol II elongation complex structure clearly show that Sen1 does not engage with Spt4/5. This implies that the Sen1 (a helicase) mediated Pol II termination mechanism is different from the one used by Rat1-Rai1 and the Integrator complexes (both possess nucleases). In addition, the study from the Pugh lab has shown that both Nrd1 and Spt5 co-occupy genomic regions both at the 5' and 3'-end of genes (PMID: 33692541). And the function of Nrd1 along with Nab3 in recruiting Sen1 is well known. Together, this indicates that Spt4/5 in yeast will engage with Sen1-bound Pol II complexes.